

# Tower-based C-band radar measurements of an alpine snowpack

Isis Brangers[1], Hans-Peter Marshall[2], Gabrielle De Lannoy[1], Devon Dunmire[1], Christian Mätzler[4], and Hans Lievens[3,1]

[1]Department of Earth and Environmental Sciences, KU Leuven, Heverlee, Belgium
[2]Department of Geosciences, Boise State University, Boise, ID, USA
[3]Department of Environment, Ghent University, Ghent, Belgium
[4]GAMMA Remote Sensing, Gümligen, Switzerland

**Correspondence:** Isis Brangers (isis.brangers@kuleuven.be)

**Abstract.** To better understand the interactions between C-band radar waves and snow, a tower-based experiment was set up in the Idaho Rocky Mountains for the period of 2021-2023. The experiment objective was to improve understanding of the sensitivity of Sentinel-1 C-band backscatter radar signals to snow. The data were collected in the time domain to measure the backscatter profile from the various snowpack and ground surface layers. The data show that scattering is present throughout the snow volume, although it is limited for low snow densities. Contrasting layer interfaces, ice features and metamorphic snow can have considerable impact on the backscatter signal. During snow melt periods, wet snow absorbs the signal and the soil backscatter becomes negligible. A comparison of the vertically integrated tower radar data with Sentinel-1 data shows that both systems have a similar temporal behavior, and both feature an increase in backscatter during the dry snow period in 2021-2022, even during weeks of nearly constant snow depth, likely due to morphological changes in the snowpack. The results demonstrate that C-band radar is sensitive to the dominant seasonal patterns in snow accumulation, but that changes in microstructure, stratigraphy, melt-freeze cycles, and snow wetness may complicate satellite-based snow depth retrievals.

## 1 Introduction

The storage of water in snow has an important impact on the water balance in many regions, as 1/6[th] of the global population depends on snow for water resources (Barnett et al., 2005). Information on the amount and distribution of snow is needed for hydrological forecasting and for making informed water management decisions. Nevertheless, there is currently no satellite, nor combination of satellites, that is capable of providing frequent, consistent, high resolution snow mass estimates in all snow conditions and terrain.

The first large-scale satellite-based snow mass estimates were derived from passive microwave sensors, with estimates going back as far as 1979 (Kunzi et al., 1982). The Chang et al. (1987) algorithm uses brightness temperature observations in the 19 and 37 Ghz frequency bands (Ku and Ka) of the Scanning Multichannel Microwave Radiometer (SMMR) to derive snow water equivalent (SWE) values. For the GlobSnow dataset, a similar approach has been applied to observations from the SMMR Special Sensor Microwave/Imager (SSM/I) and Special Sensor Microwave Imager/Sounder (SSMIS), combined with in-situ measurements and the Helsinki University of Technology (HUT) snow emission model to derive SWE over the entire



Northern Hemisphere land surface, excluding mountain ranges (Luojus et al., 2021; Takala et al., 2011; Lemmetyinen et al.,
2015). Since 2002, the Advanced Microwave Scanning Radiometer for EOS (AMSR-E) and its follow up AMSR-2 instrument
provide stand-alone snow depth (SD) estimates using a similar methodology, and also utilize the 10 GHz (X-band) channel
(Kelly, 2009). However, these passive microwave SD and SWE products have constraints. The retrievals have a relatively high
uncertainty, a low spatial resolution of around 25 km and a tendency to saturate at SWE values of about 150 mm (Luojus et al.,
2021; Mortimer et al., 2019). These challenges limit their value for operational use, especially in alpine regions.

The use of active microwave sensors at Ku-band, X-band, or a combination of both, has been studied extensively (e.g. Shi,
2006; Yueh et al., 2009; King et al., 2013). A future Ku- and X-band Synthetic Aperture Radar (SAR) mission could potentially
provide SWE estimates at higher resolutions than the existing passive microwave sensors (Tsang et al., 2022; Rott et al.,
2010). Airborne Polarimetric Scatterometer (POLSCAT) data collected in Colorado across 5 flights at Ku-band has indicated
an increase in backscatter, especially at cross-polarization, due to the increase in volume scattering with snow accumulation
(Yueh et al., 2009). Similarly as for the passive microwave sensors, some limitations exist, including the confounding impact
of snow microstructure on the backscatter sensitivity to snow (Tsang et al., 2006; King et al., 2015). These limitations were
demonstrated by Ku- and X-band measurements from SnowSAR, collected during 2 subsequent flight days, that showed limited
sensitivity to SWE in a tundra environment (King et al., 2018). Additionally, ground based radar experiments with ESA's
SnowScat during 3 months at Weissfluhjoch in Switzerland and 4 winter seasons in Sodankylä in Northern Finland showed
a complex signal at X- and Ku-band. Most of the variability of the signal was found to be related to changes in stratigraphy,
rather than SWE or SD (Lin et al., 2016). Kendra and Ulaby (1998) evaluated model and experimental C- and X-band data of
artificial snow and found increasing backscatter with SD at both frequencies. On the other hand, a tower-based radar experiment
by Strozzi and Mätzler (1998) found a decrease in C-band backscatter, but an increase at Ku-band. According to Dozier and
Shi (2000), accumulating snow may lead to increases or decreases in C-band backscatter depending on the strength of the
ground scattering. Some of the aforementioned tower based papers use tomographic profiling to characterize layering within
the snowpack (Lin et al., 2016; Frey et al., 2018). However, side-looking time-domain profiles to help understand scattering
processes have not been previously studied. This alternative approach could provide more insight into the contributing sources
to the total integrated backscatter as measured by satellites.

Another way to study interactions between radar waves and natural surfaces is through radiative transfer models (RTMs).
The earliest RTMs simulate the backscatter as the sum of the return of individual spherical snow particles (Chang et al., 1976).
In reality, the non-spherical grains in a snowpack form more complex aggregates. To account for the interactions between
densely packed particles, Tsang et al. (2006) developed the dense media radiative transfer model (DMRT). After tuning the
parameters, the model corresponded well to experimental radar measurements from QuikSCAT. DMRT simulations show
substantial amounts of cross-polarized backscatter, originating from the asymmetric structure of the aggregated particles, and
show an expected increase of Ku- and X-band backscatter with snow depth (Xu et al., 2012). Picard et al. (2018) developed the
Snow Microwave Radiative Transfer (SMRT) model, in which different modelling approaches can be combined in a modular
structure to study the scattering impacts governed by snow characteristics. Features like ice lenses, or changes in stratigraphy





are known to impact the microwave response (Yueh et al., 2009; Xu et al., 2012), but are rarely considered in RTMs. These features complicate the SD and SWE retrievals from satellite based measurements.

Although previously mentioned tower-based radar studies did not reach a consensus on the potential of C-band backscatter for SD or SWE retrieval, the launch of the ESA and Copernicus Sentinel-1 (S1) satellite mission created a renewed interest to explore the capability of global, high-resolution, multi-polarized backscatter observations at C-band. Lievens et al. (2019) proposed S1 SD retrievals based on an empirical change detection method, relating changes in backscatter to the accumulation or ablation of snow, and demonstrated the retrieval performance over mountain ranges in the Northern Hemisphere. The method

was further improved by Lievens et al. (2022) in a case study over the European Alps. A strong correlation between the S1 cross-polarization ratio (i.e., the ratio of cross- over co-polarized backscatter) and SD was also confirmed by Feng et al. (2021) and Daudt et al. (2023), the latter of which successfully developed a S1-based convolutional neural network approach for SD retrieval. Alfieri et al. (2022); Girotto et al. (2024) and Brangers et al. (2023) used the S1-based SD retrievals in data assimilation approaches covering (part of) the Alps, obtaining improved SD and discharge estimates. Notwithstanding

these advancements, studies addressing the physical understanding of C-band backscatter mechanisms in snow to support the retrieval algorithm concept are still lacking.

    This study aims to investigate the interactions between a natural alpine snowpack and C-band radar waves. The radar response will be studied in the time domain, as opposed to previous studies in the frequency domain. Using the time domain response allows us to evaluate from where in the snowpack the reflections originate, and how this evolves throughout the snow

season, governed by changes in snow properties. Detailed observations from snowpits are used to analyze the impact of snow stratigraphy on the radar signals. Furthermore, the tower radar signals are integrated and compared with S1 data to assess the potential and pitfalls of S1-based SD retrievals. The site description and radar specifications are given in Section 2. The results, discussion and conclusions are presented in Section 3 and 4.

## 2 Materials and Methods

### 2.1 Study site


The radar was installed at a site in the Idaho Rocky Mountains, just outside the boundaries of a local ski resort called Bogus Basin (43.76°N, 116.09°W, see Figure 1). The site is maintained by the Cryosphere Geophysics and Remote Sensing (CryoGARS) group at Boise State University and has accommodated several other research projects, such as the National Aeronautics and Space Administration (NASA) SnowEx campaign during the winters of 2019-2020 and 2020-2021 (Marshall et al., 2020).

The vegetation at the site consists of shrubs and grasses, with groups of conifer trees in the surrounding area (Figure 1c). Continuous observations of weather variables such as precipitation and air temperature, and snow variables such as SD and SWE are available from a nearby snow telemetry (SNOTEL) site (ID 978). The site lies on a flat section of a South-East facing slope, at an elevation of 1930 m. The sandy loam upper ground layer typically remains unfrozen at constant temperature of 0°C during the snow season. Based on visual estimates from a vegetation free location during the summer, the soil root mean square

(RMS) height was estimated as 1 cm, which makes it rather smooth relative to the 5.5 cm wavelength. The Bogus Basin site is





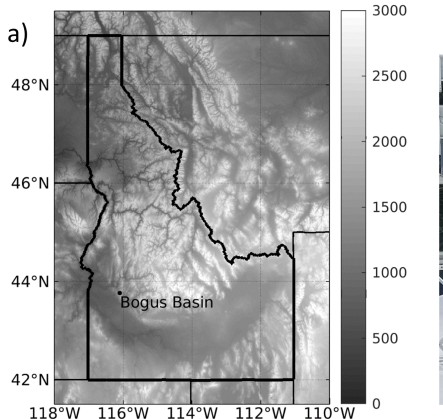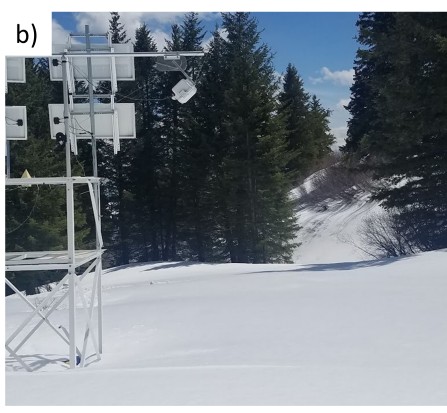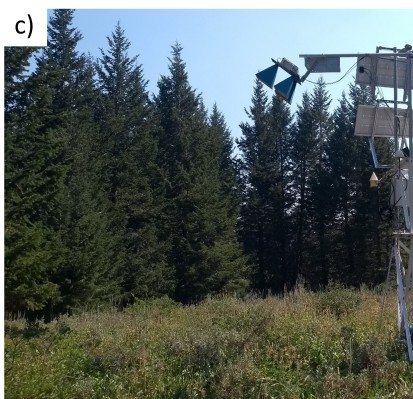

**Figure 1.** Study site. (a) The location of the site in Idaho (US), with the background representing the elevation (m) from the Shuttle Radar Topography Mission (Farr et al., 2007). (b) The site in winter with the box containing the radar with tapered slot antennas as used in winter 2021-2022. (c) The site and its vegetation in summer during installation of the radar with horn antennas used in winter 2022-2023.

characterised by a Montane Forest snow climate (Sturm and Liston, 2021), with relatively warm and moist snow conditions, and a possibility for mid-winter rain-on-snow events and melt-freeze cycles. The median peak SWE (1991-2020) is ∼650 mm, according to the SNOTEL measurements.

## 2.2 Radar instrument

A custom-built impulse radar was designed for this experiment. It has two receive and two transmit channels, with two horizontally (H) and two vertically (V) polarized antennas attached. This allows the instrument to capture all four polarizations, i.e., HH, VV, HV and VH, where the first and second letter indicate the polarization of the transmit and receive channel, respectively.

The radar was attached to a metal tower (Figure 1b-c) and was mounted approximately 3.6 m above the soil surface in winter 100 2021-2022 and 4.0 m during winter 2022-2023. The instrument had a look angle of 40°, comparable to the Sentinel-1 incidence angles (29.1°-46.0°), although S1 captures a wider range of local incidence angles in complex terrain due to topography. Figure 2 illustrates the radar configuration. The two ellipses indicate the instantaneous footprint along the time-domain profile, and the footprint as projected onto the ground surface. The area along the time-domain profile has a radius r and is a function of the half power beamwidth (HPBW) and the distance from the radar (R). The area projected on the ground surface was used to 105 normalize the integrated backscatter values, independent of the snow depth.

Once per hour or per day, depending on the experiment period, the radar sends out pulses of microwave energy and subsequently records the reflections directly in the time domain. This approach is different compared to some earlier studies which focused on frequency domain measurements (Morrison et al., 2007). The signal return is repeatedly sampled with small time offsets, to build a profile through the air and snowpack. With an effective sampling frequency of 39 GHz, and assuming that



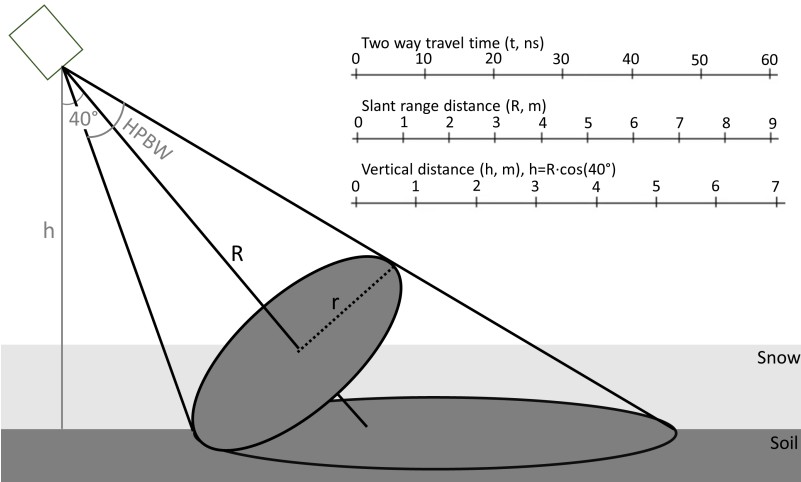

**Figure 2.** The geometry of the radar setup. The ellipses indicate the radar footprint along the time-domain profile and the footprint as projected onto the ground surface. The depicted travel time to distance conversion assumes waves travel at the speed of light.

the waves travel at the speed of light in vacuum, this corresponds to one sample every 4 mm (or 8 mm two-way, same for both seasons). With a total sample size of 3072, the measured slant range is ~12 m (24 m two-way). For each measurement, multiple pulses are sent out and the returning signals are averaged to limit the noise level. To ensure good range resolution, the radar has an ultra-wide bandwidth. The Gaussian shaped pulse is around 1 ns long and centered around 5.4 GHz, or at a wavelength of 5.5 cm, with a (-10 dB) bandwidth of 2.4 GHz.

Two types of antennas were investigated. During the winter season of 2021-2022, four linear tapered slot antennas (LTSA) were used. These consist of a 2D V-shaped horn made of conducting material on a simple circuit board. The antennas were relatively small, making them practical to install. Despite the flat design, the antennas produce an almost symmetrical beam. The radar and antennas were placed within a sturdy, watertight plastic box. Radar absorbing foam was placed in-between the antennas, at the back, and at the sides of the box to limit noise caused by radar waves bouncing between the electronics and

antennas, and by systematic reflections from the metal tower structure. In the 2022-2023 season, the LTSA antennas were replaced by 3 standard gain horn antennas, measuring HH and HV polarization only. These antennas created a much more narrow beam, resulting in a smaller footprint that simplified the interpretation.

    A downside of the horn antennas is the larger required far-field distance ($D_F$). At this distance the wavefront can be considered planar, and standard mathematical expressions can be used (Ulaby et al., 2014). The far-field distance can be calculated as

$D_F = 2d^2/\lambda$, with $d$ the longest dimension of the antenna and $\lambda$ the wavelength. Using $\lambda = 5.55$ cm and $d_{\mathrm{horn}} = 31.6$ cm, the horn antenna requires a distance of ~3.6 m from the target. In contrast, for the smaller LTSA ($d_{\mathrm{LTSA}} = 6.6$ cm), $D_F$ is only 16 cm.

    To determine the beamwidth of the radar, the antenna pattern was measured. Usually these measurements are made under controlled circumstances in an anechoic chamber, but due to budget and practical constraints, an alternative setup was used





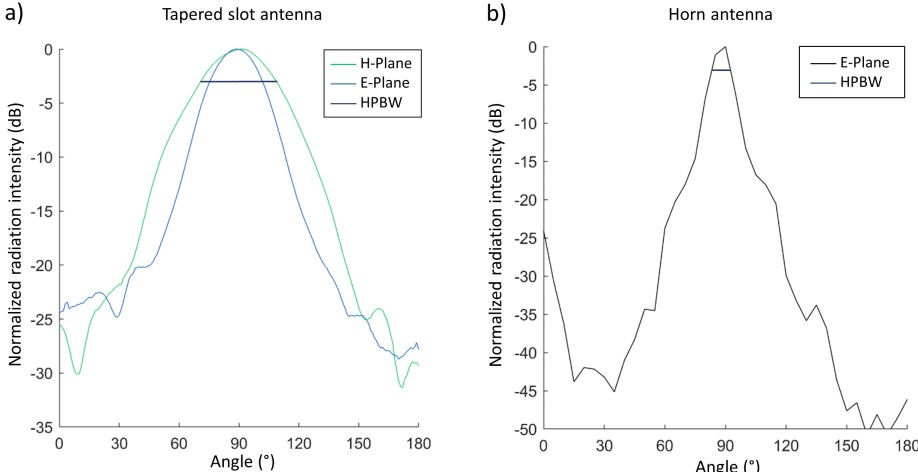

**Figure 3.** Approximate radar antenna pattern for the (a) linear tapered slot antenna and (b) horn antenna.

here. The experiments were carried out in the large open space of a warehouse, where limited reflections from nearby objects were expected. The few unwanted ambient reflections may cause a slight overestimation of the sidelobes. The antenna pattern of the radar is determined by observing a metallic sphere from a range of angles at regular intervals. In case of the LTSA, the radar was kept inside the box as used for installation in the field. The response was measured along the planes of both the electrical and magnetic field vectors (E- and H-plane). Measurements were made automatically at $1°$ steps with a pan-tilt

system for the LTSA, and manually at $\sim 5°$ steps for the horns. As can be seen in Figure 3, for the LTSA the beam is slightly elliptical with a HPBW (-3 dB) of around $40°$. For the horn antennas only the E-plane was measured, with an estimated HPBW of $10°$. Using the HPBW to estimate the footprint area, in 2021-2022, with a tower height of 3.6 m, led to an area of around $13\,\mathrm{m}^2$. In 2022-2023, with a tower height of 4.0 m but a much smaller beam, the footprint area was close to $1\,\mathrm{m}^2$.

    A minor issue was identified for the vertical receiving channel during the 2021-2022 season. As a result, the data collected

in VV polarization is slightly noisier than for the other channels. The radar was temporarily taken down during the site visit on December 17, 2021, to troubleshoot this issue, causing a minor discontinuity in the timeseries.

### 2.3 Radar signal processing

After the raw radar data collection, some processing steps are necessary to de-noise the signal and to derive functional data for the interpretation of the radar-snow interactions. The raw data consists of traces in the time domain for all measured polar-

izations. Each radar measurement is a combination of multiple pulses that were already integrated within the radar software to limit noise. The raw signal is saved in terms of analog to digital converter (ADC) counts. These ADC counts, however, depend on user settings such as the amount of iterations, and were normalized into a voltage as explained in the instrument manual (FlatEarth, 2016).

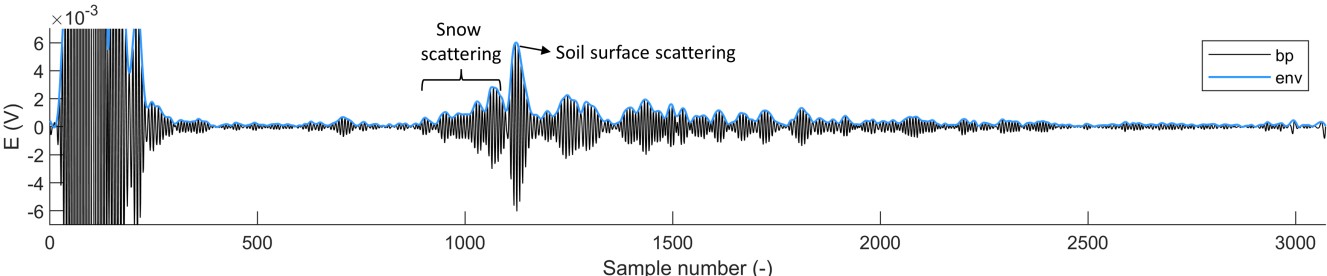

**Figure 4.** Illustration of a single radar trace taken on December 15th 2021 in HH polarization with snow on the ground surface. The raw data is plotted after applying a bandpass filter (bp, black line) and taking the envelope (env, blue line).

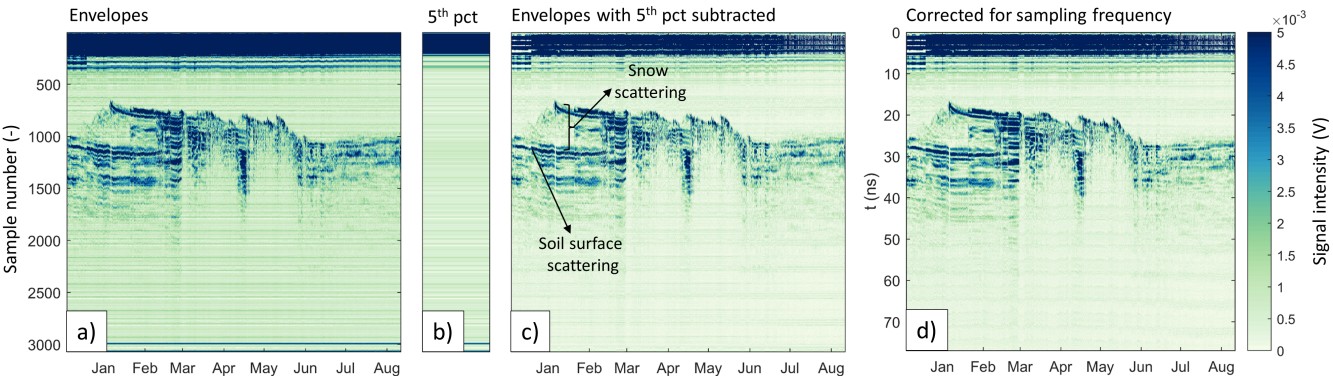

**Figure 5.** Illustration of further data processing steps, on the data of 2021-2022. The y-axis shows the sample number which increases with distance from the radar, the x-axis the time throughout the season. In panel a) the envelopes of the entire season are plotted for the VV channel, b) shows the 5th percentile of each sample/row, c) contains the cleaned data after subtracting b) from a), and d) shows the data after a sampling frequency correction with travel time on the y-axis.

An example of a time domain backscatter measurement (1 trace) is shown in Figure 4. A bandpass filter (5.4 ± 3 GHz) was
applied to remove low frequency noise. The first part of the signal is very strong and is caused by the bounces between the electronics and antennas. The main reflection from the ground surface is located around sample 1100. The oscillating wave is turned into a smooth signal by taking the envelope, i.e. the curve outlining the maximum values along the measurement time window. When plotting all the enveloped traces at each timestep in a single plot, variations in the signal strength, along with the accumulation and ablation of the snowpack can be observed (Figure 5a). Every column in the figure represents one hour
of the snow season and the sample numbers on the y-axis increase with distance from the radar. The ground surface reflection is visible during the accumulation season (beginning around sample number 1050 in January), but becomes invisible during the wet snow season. As snow accumulates, the ground surface is observed at increasing sample numbers (it appears to move further away from the top of the plot), which is caused by a slower wave propagation in snow than in air.



A common processing step in Ground Penetrating Radar (GPR) data is to subtract the mean of all traces to remove consistent
noise. In our case, with a relatively constant ground location, this could decrease the estimated ground contribution. As an
alternative, the 5th percentile of the data in time was calculated for each sample number (Figure 5b) and subtracted from the
envelopes (Figure 5c).

Next, the sample numbers ($s$, -) were converted to travel times ($t$, ns) using the sampling frequency ($f_s = 39\,\text{GHz}$): $t = s/f_s$.
Specifically for our instrument, depending on the temperature, the radar recorded samples at different speeds. The variable
sampling frequency was recorded for each measurement. Since the number of samples stays constant, this means a slightly
shorter or longer time frame can be recorded. For example, during the season of 2021-2022 at the Bogus Basin site, the
minimum and maximum recording times were 75.7 and 81.3 nano seconds, corresponding to distances of 11.35 and 12.19 m,
respectively, assuming the waves traveled at speed of light in vacuum. All traces were resampled to match the length of the
trace with the longest recording time. The total energy per trace was conserved. The resulting sampling frequency corrected
data is shown in Figure 5d, where strong reflecting layers like the ground signal appear more continuous.

The time axis can be converted into slant range distance from the radar by assuming waves travel at the speed of light in
vaccuum ($c$, see Figure 2). However, this is not a perfect conversion, since as the snow accumulates, waves travel more slowly
and the ground reflection appears slightly later on the time-axis. Additionally, the angle of wave propagation gets redirected
when transitioning into the snowpack due to refraction. The impact of the slower travel speed ($v_{\text{snow}}$) along with the refraction
effect can be taken into account. Both these effects depend on the snow permittivity ($\epsilon_{\text{snow}}$) in the following way:

$$\sqrt{\epsilon'_{\text{snow}}} = \frac{c}{v_{\text{snow}}} = \frac{\sin(\theta_{\text{inc}})}{\sin(\theta_{\text{snow}})} \tag{1}$$

in which $\theta_{\text{inc}}$ and $\theta_{\text{snow}}$ are the angles relative to the normal at which the waves travel in the air and snow respectively. The
real part of the dielectric constant of the snowpack depends on the snow density ($\rho_{\text{snow}}$, g/cm$^3$). Based on Hallikainen et al.
(1986), we estimated the permittivity as $\epsilon'_{\text{snow}} = 1 + 1.83\rho_{\text{snow}}$. The snow density was calculated at each timesetp from the
SNOTEL measurements of SWE and SD and was considered to be vertically uniform throughout the snowpack. The travel
time was converted to slant range distance and vertical distance similar to the approach of Strozzi (1996).

The signal obtained from these tower experiments is the received electric field strength ($E$, Volt) in time. To compare to
satellite or other tower based measurements, this signal must be converted into a single integrated normalized radar cross
section, $\sigma_0$ (-), based on the radar equation:

$$\sigma_0 = \frac{P_{\text{rec}}}{P_{\text{tr}}} \cdot R^4 \cdot \frac{(4\pi)^3}{G_{\text{rec}}G_{\text{tr}}\lambda^2} \cdot \frac{1}{A} \tag{2}$$

with subscripts 'tr' and 'rec' denoting transmitted and received, $P$ the power (W), $G$ the antenna gain (-), $R$ the slant range
distance from the radar (m), $\lambda$ the wavelength (m), and $A$ the footprint area (m$^2$). The total received power $P$ is proportional
to $E^2$; $P = E^2/(2\eta)$, with $\eta$ (W/V$^2$) the impedance (Ulaby et al., 2014). The snowpack and the underlying soil surface are
distributed targets, the value $P_{\text{rec}}$ contains the contributions of all the scatterers within the footprint. The target acts as a cloud
of many scatterers, each of these is contributing to the signal depending on their distance from the center of the beam.

Due to the limited height of the tower, the footprint area is an ellipse when projected on the Earth (see Figure 2), and is
proportional to the squared distance from the radar. Along the time domain profile, the area contributing to a sample can be



approximated as a conical cap or circle of area $A = \pi r^2$, with $r$ the radius of the footprint, calculated as $r = R \cdot \tan(\text{HPBW}/2)$. The instantaneous illuminated area is thus also a function of the squared distance from the radar.

Substituting the area in the radar equation and aggregating the constants gives:

$$\sigma_0 = E_{\text{rec}}^2 R^2 k \tag{3}$$

in which $P_{\text{tr}}$, $G_{\text{rec}}$, $G_{\text{tr}}$, $\eta$, $\lambda$ and HPBW are considered constant and can therefore be aggregated into a single constant $k$ (1/Wm$^2$). The value of $k$ depends on the radar system (e.g., the antenna characteristics, radar frequency, the connecting cables, tightness of the cable connections, etc.) and will vary between each of the channels. Its value is determined by a calibration

based on targets with a known radar cross section.

In this experiment, $E_{\text{rec}}$ is split over multiple small bins along the time domain profile. For a single time bin (with sample number $(i)$), the received signal $(s)$ can be written as:

$$s(i) = E_{\text{rec}}^2(i)R^2(i) \tag{4}$$

Note that the return in each time domain bin is multiplied by $R^2$. For a satellite or airplane at considerable altitude, $R^2$ can

be considered constant throughout the measurement of a pixel. For this tower based experiment, however, $R^2$ varies throughout a radar trace and is estimated from the wave travel time. The wave velocity and refraction are corrected for using the snow density. Conceptually, this $R^2$ correction is needed because with distance, the energy spreads out over a larger surface. The further the signal has to travel the lower the captured power at the receiving antenna will be. Contrary to a satellite view of the Earth surface, the footprint area of a tower based radar changes substantially throughout the time domain measurement.

The multiplication with $R^2$ takes this variable footprint into account and allows us to compare the relative contributions of snowpack layers at different depths, similar as how they would contribute to the received satellite signal.

The time domain signal was integrated and calibrated to allow for comparisons with previous tower- and satellite-based radar measurements. Each tower radar trace consists of a profile of multiple consecutive time-bins (of length $\Delta t$) or samples $(i$; the number of samples is $N)$. To calculate the integrated signal (per hourly trace), the return per bin was summed into a

single value $(S$, W/m$^2)$, where the noise band at the top of the profile was omitted (i.e. top 500 samples):

$$S = \sum_{i=1}^{N} E_{\text{rec}}^2(i) \cdot R^2(i) \cdot \Delta t \tag{5}$$

Then, the radar return $S$ can be calibrated by multiplication with constant $k$ (see Eq. 2.3). Sarabandi et al. (1990) use a method for quad-pol radar systems that requires the measurement of two targets, where the scattering matrix needs to be known for only one of the targets. Similarly, here, the co-pol channels were calibrated with a metal sphere with known scattering matrix

because of its relatively simple implementation in the field. Considering that the theoretical $\sigma_0$ from calibration targets as mentioned in literature refer to the total received power (Pancera et al., 2010), the calibration was applied to the integrated backscatter values and not to the traces in the time domain.



Following Sarabandi et al. (1990), the co-polarized channels can be calibrated based on the measurement of the sphere as follows:

$$\sigma_{0,\mathrm{VV}} = S_{\mathrm{VV}} \cdot \frac{\sigma_0^{\mathrm{cal}}}{S_{\mathrm{VV}}^{\mathrm{cal}}} \tag{6}$$

where quantities with the superscript 'cal' refer to those obtained with the calibration sphere. The integration of $S_{\mathrm{VV}}^{\mathrm{cal}}$ (based on Eq. 5) was applied only to those samples where the sphere was located, to limit the impact of background noise. The theoretical $\sigma_0^{\mathrm{cal}}$ value for co-pol was calculated from the sphere radius ($r_{\mathrm{cal}}$): $\sigma_0^{\mathrm{cal}} = \pi r_{\mathrm{cal}}^2 / A$. In our case $r_{\mathrm{cal}}$ is 12 inch (30.48 cm), resulting in $\sigma_0^{\mathrm{cal}} = 0.29/A$ for co-polarization, with $A$=13 m$^2$ for the LTSA and $A$=1 m$^2$ for the horn antenna setup. Scattering from a sphere in cross-polarization is negligible.

For the cross-polarized channels, as a target with strong cross polarization (quantities with superscript x) the return of a metal mesh was measured. According to the theorem of reciprocity, the $\sigma_{0,\mathrm{VH}}$ and $\sigma_{0,\mathrm{HV}}$ should always be equal. The return was converted into a normalized radar cross section based on the methodology of Sarabandi et al. (1990):

$$\sigma_{0,\mathrm{VH}} = S_{\mathrm{VH}} \cdot \sigma_0^{\mathrm{cal}} \sqrt{\frac{S_{\mathrm{HV}}^{\mathrm{x}}}{S_{\mathrm{VV}}^{\mathrm{cal}} S_{\mathrm{HH}}^{\mathrm{cal}} S_{\mathrm{VH}}^{\mathrm{x}}}} \tag{7}$$

However, we were not able to apply the full calibration as foreseen. In the 2021-2022 season, the noise level of the HV channels was too high, and during the 2022-2023 season only one cross-pol channel was available. $S_{\mathrm{HV}}^{\mathrm{x}}/S_{\mathrm{VH}}^{\mathrm{x}}$ was therefore considered to be 1.

The resulting integrated and calibrated backscatter is shown in Figure 9. As expected, the signal of the cross-polarized channels is generally lower than that for the co-polarized channels. The same calibration constants applied to the integrated return (i.e. $\sigma_0^{\mathrm{cal}}/S_{\mathrm{VV}}^{\mathrm{cal}}$ and $\sigma_0^{\mathrm{cal}} \cdot \sqrt{S_{\mathrm{HV}}^{\mathrm{x}}/S_{\mathrm{VV}}^{cal} S_{\mathrm{HH}}^{cal} S_{\mathrm{VH}}^{\mathrm{x}}}$) were multiplied with the time domain signals from Figure 6 so that the relative strength of the channels can be compared in the time domain figures. However, it is important to note that the colorbars from Figure 6 do not represent absolute backscatter values. The absolute calibration is only valid for the integrated signals.

## 2.4 Ground-based reference data

Near the site ($< 1$ km away), a SNOTEL station is present (ID 978). This station continuously measures snow variables like SWE and SD, weather variables like precipitation, air temperature and solar radiation and soil variables like soil moisture (volume fraction of liquid water) and soil temperature. During the accumulation season SD and SWE are highly correlated and very similar between the nearby SNOTEL and radar sites. At the radar site, detailed snowpack properties were collected from snowpits during 10 site visits throughout the 2021-2022 season. Specifically, the site was visited on December 17th, January 1st, 6th, 13th and 17th, February 1st, March 2nd, 17th, and 26, and May 12th. These snowpit measurements included snow depth, profiles of density, temperature and stratigraphy, and manual observations of hardness, wetness, grain size and grain shape for each observed snow layer. The density profiles were measured in 10 cm increments using a wedge shaped cutter of 1 dm$^3$. Grain sizes and shapes were visually estimated on a card with a reference grid and with a microscope. During the 2022-2023 winter, no regular site visits were made and thus the reference data is limited to the SNOTEL measurements for this season.



## 2.5 Comparison of tower- and S1-based radar data

Sentinel-1 (S1) is a near polar-orbiting SAR mission from the ESA and Copernicus program. It consists of two identical satellites, S1 A and B, that follow the same 12-day repeat cycle. From December 2021 onward only observations from S1A are available due to a sensor failure on the B satellite (ESA Sentinel-1 Team, 2022). The onboard C-band SAR instrument measures at a spatial resolution of 5x20 m$^2$. For this project and to limit the speckle noise, the data was upscaled to 100x100 m$^2$ through multi-looking. S1 data was downloaded from the Alaska Satellite Facility and processed using ESA's Sentinel Application Platform (SNAP) toolbox following the methodology of Lievens et al. (2022). More specifically, we applied the precise orbit files, border noise removal, thermal noise removal, a radiometric calibration and lastly a terrain flattening to gamma ($\gamma_0$) (Small, 2011). Here, we used terrain flattened gamma ($\gamma_0$) instead of the often used sigma ($\sigma_0$), since the $\gamma_0$ algorithm more adequately corrects for the effects of terrain variations (Small, 2011). For the comparison between S1 and the tower radar data, we are mostly interested in the temporal dynamics rather than the absolute backscatter values. This makes the decision between $\gamma_0$ or $\sigma_0$ of lesser importance.

The study site is covered regularly by 4 different relative orbits, two of them ascending (A) with passover times around 6 pm local time, and two descending (D) with passover times around 6 am. Each orbit measures the site from a different look angle. Specifically, the local incidence angles are 25°, 36°, 38° and 47° for orbits 144D, 71D, 93A and 166A respectively. To compare these different orbits, the backscatter data were rescaled by rescaling the mean of each orbit to the overall pixel mean (across all orbits). The tower based radar profiles were integrated into a single value as explained in Section 2.3.

## 3 Results and Discussion

### 3.1 Interpretation of time-domain reflections

**2021-2022 season**

The hourly LSTA backscatter data of 2021-2022 and the measurements from the nearby SNOTEL station are presented in Figure 6. The ground surface is visible at 3.6 m in the radar plots. During the snow accumulation season, from December till the beginning of March, the backscatter steadily increases over time. When the snow gets wet, the return decreases. The backscatter evolution in time is mostly similar for each polarization. The cross-pol signal is generally about 5 dB lower than the co-pol signal. However, it must be kept in mind that the values portrayed in the colorbar are not absolute backscatter values. An absolute calibration was applied only to the integrated values (see later, Figure 9).

Figures 6 a-c show the steady accumulation of snow until the beginning of January, with some air-snow interface, snow layer interface and snow volume scattering happening throughout the entire dry snow period. The fact that we can distinguish the air-snow interface at the beginning of the accumulation season, when snow densities are low, indicates that there is some volume scattering present in the snowpack. The relative strength of snow versus ground scattering changes throughout the dry



**Figure 6.** Hourly tower radar data (dB) throughout the winter season 2021-2022 at the Bogus Basin research site, collected with the LTSA. The upper plots show the tower radar data (dB) in (a) VV, (b) HH and (c) VH polarization. The values on the colorbar do not represent absolute backscatter values. The y-axis represents the vertical distance from the radar. The lower plots show recordings from the nearby SNOTEL stations of (d) SD and SWE, (e) 2 m air temperature and (f) volumetric soil moisture fraction and soil temperature at 2 inch depth.



snow season with stratigraphy and microstructural variations. First, scattering from the ground is dominant. However, at the end of the dry snow period, scattering within the snowpack has become the main contribution to the backscatter signal.

In March, the melt season has started. When the snowpack is wet, we typically see strong reflections at the snow surface due to the increase in dielectric contrast between the air and the wet snow surface, and almost no reflections afterwards (from deeper layers) due to the strong absorption of the signal by wet snow. The strong reflections from the snow-ground interface disappear. When the snowpack (partly) refreezes (e.g. between the 4th and 14th of March or in mid April), scattering within the snowpack again becomes apparent from the signal. This strong impact of liquid water on the scattering response was expected and agrees with previous findings (Nagler et al., 2016; Lund et al., 2022; Marin et al., 2019). The wet snow can thus be easily discerned from the dry snow, confirming the potential of C-band active microwave observations for monitoring wet snow. Furthermore, there are also diurnal changes in the water content. For instance, January 6th, temperatures rose above freezing, and the top of the snowpack started melting. In Figure 6, this short melt period can be recognized by a brighter vertical stripe in the figure that contrasts with the scattering from the surrounding days. This melt event resulted in the appearance of a crust at the top of the snowpack, a feature that exists throughout the remainder of the dry snow period. Striping from melt-freeze cycles can also be noticed mid February, where occasional surface melt events appeared around noon; or during the wet-snow period, when the snowpack partly refroze during the nights. This diurnal variation highlights the impact of the satellite passover time on backscatter values, which will be further discussed in Section 3.3.

The depicted snow year had anomalously high accumulation during December, followed by two months with almost no new snowfall. Although the SWE and SD remained relatively constant during January and February (Figure 6d), the stratigraphy and/or microstructure of the snowpack did change. Correspondingly, the scattering properties of the snowpack evolve over this period. To illustrate this, a representation of the stratigraphy taken from snowpits during the season is shown in Figure 7. At the bottom of the snowpack, there was a crust from early season snowfall that partly melted and refroze. The disintegration of this crust can also be seen in Figure 6. During December there was heavy snowfall, leading to a low density snowpack of small, decomposing grains. In January, two melt events occurred that are also apparent in the radar signal. On the 6th, a solid ice crust formed near the surface. During the site visit on the 24th of January, more evidence of melt was visible in the snowpack: the appearance of another thin melt-freeze crust near the top and several ice clumps. At this time, the grains were also more rounded. At the beginning of March, the snowpack was wet and isothermal with percolation features throughout the pack.

Changes in radar backscatter occurred during periods of nearly constant SD and SWE due to changes in stratigraphy, highlighting a potential source of uncertainty in the estimation of SD from backscatter increases. Based on the above results, the Lievens et al. (2022) methodology is expected to work best in the dry snow period, before significant melt-freeze cycles occur. The different polarizations are found to have similar scattering features, although the strength of the cross-pol signal is consistently lower than the co-pol signal (note the different colorbar). When the snowpack becomes more or less transparent, it does so similarly in all channels. Lievens et al. (2022) also noticed how at some sites the S1 VV and VH increased similarly with SD, whereas at other sites, VV stayed mostly constant and only VH increased with snow accumulation.

The radar figures additionally show evidence of multiple scattering, i.e. the radar signal reflecting multiple times within the snowpack, between its layers, or at the ground/snow interface. These multiple reflections have longer travel times than single





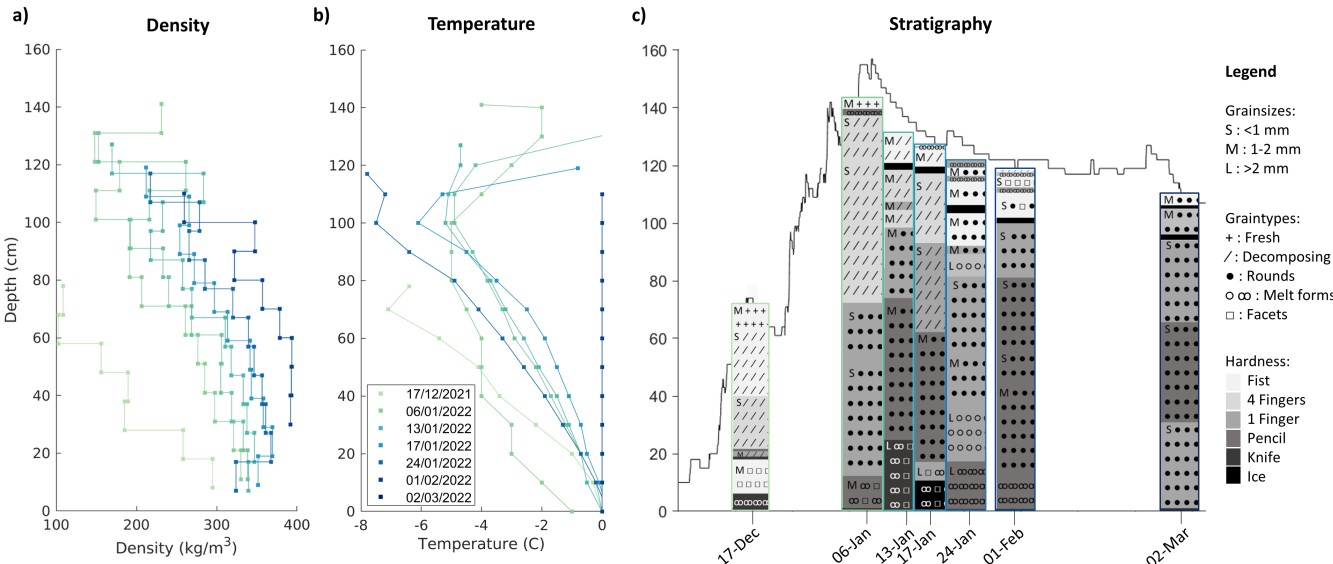

**Figure 7.** Snow pits taken near the Bogus Basin site during the winter 2021-2022. The observed snow properties are shown throughout the profile: a) density (kg/m$^3$), measured with a 10 cm density cutter, b) temperature ($^\circ$C) measurements at 10 cm intervals and c) stratigraphy.

reflections and can therefore appear below the main ground reflection, as can be seen in Figure 6. For instance, during the main dry snow accumulation period (before January 6th) the below ground scatter (between vertical distance of 3.8 and 6 m) increases with snow accumulation. With the development of snow and ice layers, some ice crusts appear to be mirrored below the ground, which can potentially be explained by multiple-scattering between the crust(s) and the ground.

As soon as there is snow on the ground, the soil temperature stays consistently at 0°C. Soil moisture is also mostly constant,
except during the melting season. Therefore, soil moisture and soil temperature do not appear to substantially impact the backscatter during the dry snow season at this site. However, soil freeze-thaw events are known to influence backscatter (Naeimi et al., 2012) and might impact the backscatter measurements in the shoulder seasons. The snow-free spring backscatter signal indeed changes with soil moisture fluctuations.

**2022-2023 season**

Figure 8 shows the measurements taken during the winter of 2022-2023 with the horn antennas. The data were collected at hourly intervals until February 3rd. For the remainder of the season, due to technical issues, data are available at daily timesteps, recorded at 2 am. The SD reached a peak of 287 cm at the end of March, which was the highest SD recorded since the SNOTEL measurements started in 1999. Because of this extreme snow accumulation, the far-field distance was not respected at all times (see Section 2.2). Nevertheless, even the measurements at the peak SD appear to be stable and comparable to the remainder of
the season. Compared with the previous year, the scattering is much more homogeneous, with overall lower apparent influence of melt-freeze events. During a warm period around Christmas, scattering temporarily increased, especially at co-polarization.





The strong scattering feature that appeared gradually fades, possibly due to disintegration caused by local temperature gradients and curvature effects. Similar backscatter increases are observed in mid March and early November, likely originating from melt-freeze cycles. These changes also appear in the signal below the main ground return. An interesting feature is the apparent mirroring of snow scattering around the ground surface, which could indicate contributions from multiple scattering that require longer travel times.

Overall, low density snow at the beginning of the accumulation season is mostly transparent, while the snow volume scattering increases over the course of the dry snow season. As opposed to the previous season, the relative contribution of the ground remains stronger than the snow scattering. The onset of melt occurs in early April, after which the ground surface reflections disappear and the snow-air interface becomes the dominant scatterer. Two cold spells during April cause the snowpack to partially refreeze, temporary leading to increased snow scattering.

**Comparison of setups**

Between the two observed seasons, the backscatter signal changed slightly. For example, the ground reflections appear stronger during the 2022-2023 season than for 2021-2022. The SNOTEL soil moisture values during the dry snow period were similar during both seasons. The differences between the two seasonas can be explained by differences in the position where the radar was attached to the tower, and thus differences in the instrument footprint. Additionally, different radar antennas were used in the two seasons, which might have had unintended impacts on the collected radar measurements. In 2021-2022, linear tapered slot antennas (LTSAs) were used with a beam of $\sim 40°$, and a footprint of around $13\,\mathrm{m}^2$. In 2022-2023, these antennas were replaced with horn antennas with a beam of $\sim 10°$, and a footprint of around $1\,\mathrm{m}^2$. The smaller footprint of the horn antenna may be more susceptible to speckle effects. Previous work (e.g., Strozzi and Mätzler (1998) and Kendra and Ulaby (1998)) combined multiple independent measurements with small footprint displacements to minimize these speckle effects; however, in contrast to these previous ground-based radar studies, our measurements were made at a fixed position. The broader beamwidth slot antennas effectively act as a despeckling filter by averaging over multiple incidence angles. The beamwidth could also impact the response, as multiple reflections may be more likely to return to the beam when it is wider. The impacts of radar footprint and beamwidth on the received backscatter signal requires more research and will be the subject of future studies.

### 3.2 Snow melt state

S1 backscatter, and more specifically the relative values of the ascending and descending tracks, contains information on the snow melting state (Marin et al., 2019; Lund et al., 2022). The dataset presented here provides additional insight necessary to further understand the processes at hand. Tower-based measurements taken at 6 am and 6 pm can be used to represent the descending (D) and ascending (A) tracks, respectively. As opposed to S1 tracks which have varying incidence angles for different orbit paths, the tower-based radar data has the advantage of a stable viewing geometry. According to Marin et al. (2019), the melt season can be separated into three phases. First, during the moistening phase, the upper layer of the snowpack starts getting wet during the warmest part of the day and refreezes during the night. Since the presence of water





**Figure 8.** Tower radar data (dB) for the winter season 2022-2023 at the Bogus Basin research site. The data is available at hourly timesteps through February 3th, after which observations were made daily. The upper plots show the tower radar data (dB) in (a) co-pol and (b) cross-pol polarization. The values on the colorbar do not represent absolute backscatter values. The y-axis represents the vertical distance from the radar. The lower plots show recordings from the nearby SNOTEL stations of (c) SD and SWE, (d) 2 m air temperature and (e) volumetric soil moisture fraction and soil temperature at 2 inch depth.





varies throughout the day, the ascending tracks have a lower backscatter than the descending tracks. During the second stage of snow ripening, the snow gradually transitions to full saturation. During this stage, the snow is also wet during the morning, leading to low backscatter values in both ascending and descending overpasses. Lastly, during the runoff stage, the snowpack is fully saturated and the SWE decreases. Marin et al. (2019) mention an increase in backscatter with decreasing SWE at the end of the melt season, which they suggest is caused by changes to surface roughness and stratigraphy. The local backscatter

minima corresponds with the start of the runoff period.

     In Figure 9a, the temporally (vertically) integrated and calibrated tower based backscatter ($\sigma_0$) from the 2021-2022 season were plotted daily at 6 am and 6 pm to represent S1 descending and ascending tracks. The comparison with S1 is following below; the focus here is first on the temporal dynamics in the tower based signal. Note that the depicted snow year and site had a high amount of melt-freeze cycles that might not be representative for typical alpine snow conditions. However, the

observations are still valuable to gain insight on the impact of satellite passover time. During the dry snow period, the 6 am and 6 pm $\sigma_0$ values were equal and slightly increasing over time. During the January 6th melt event, $\sigma_0$ was low during the morning and evening measurements, but returned to normal dry snow values the following day. In mid-February, several small melt and refreeze cycles led to lower afternoon values. After this, the snowpack remained frozen until a more intense melt event in the beginning of March. During most of March, the backscatter remained low for both the morning and evening measurements,

suggesting the snowpack was in the ripening phase, as characterized by Marin et al. (2019). Around mid-April temperatures dropped and the snowpack temporarily refroze, after which melt restarts. At the end of April and beginning of May, the backscatter was consistently low. This corresponds to a period of runoff where SWE decreases and soil moisture increases. The final melt stage started on May 14th, when the backscatter reaches a local minimum. From then on, the backscatter increased until the snow cover disappears.

For the 2022-2023 season (Figure 9b), tower based $\sigma_0$ was available at daily timesteps at 2 am from February 3th onward. Unfortunately, this includes the melt-onset period when the comparison of 6 am and 6 pm signals would be most interesting. The hourly data collected before February 3rd shows the sub-daily $\sigma_0$ variability. When the snowpack stays dry, $\sigma_0$ was relatively constant throughout the day. The most drastic decreases correspond to melt events.

     For both seasons, our data shows that for the multiple observed melt-freeze cycles, the local backscatter minima correspond

quite well with the start of runoff, where SWE starts to decrease and soil moisture increases (gray shading in Figure 9). The tower-based data confirms the strong sensitivity of radar backscatter to snow wetness. When the snow is wet, the ground contribution can be neglected and the snow surface becomes the dominant scatterer. For the first observed season, multiple melt-freeze cycles followed each other closely. Nevertheless, generally, the different snow melt stages described by Marin et al. (2019) can be determined in the collected tower-based data. Infrequent S1 observations from different orbits and viewing geometries, can however lead to a potential mislabeling of these wetness stages. For example, during the cold period in mid

April, both SWE and SD increased, however due to limited S1 observations, this period could be marked as runoff.



**Figure 9.** Integrated tower radar signals (dB). a) Measurements from Bogus Basin in 2021-2022 made at 6 am and 6 pm for VV and VH polarization, compared to Sentinel-1 backscatter from the site pixel. b) shows the responses from the same site in 2022-2023, at hourly timesteps until February 3th and for the remainder of the season daily measurements at 2 am. Melt events are indicated and the gray shading marks moments were SWE starts to decrease. To merge different S1 viewing geometries, the orbital $\gamma_0$ values were rescaled to the overall pixel mean $\sigma_0$. The incidence angles from the S1 orbits are $25°$, $36°$ $38°$ and $47°$ for for orbits 144D, 71D, 93A and 166A respectively.



## 3.3 Comparison to Sentinel-1

Figure 9 also shows the S1 $\gamma_0$ as symbols on top of the lines representing the tower-based backscatter. Even after the calibration described in Section 2.3, the range of the tower-based $\sigma_0$ differs from S1 $\gamma_0$. Some differences are expected since the tower radar footprint is flat, and contains only low vegetation. In contrast, S1 pixels are larger and contain rougher, more variable terrain and vegetation coverage. Moreover, S1 observes the site from 4 different orbits with variable viewing geometries. Since we are mostly interested in the temporal dynamics, the similar trends in the S1 and tower backscatter signals justify their comparison. The correlation between the S1 and tower backscatter is 0.40 for the dry snow season or 0.39 for the entire snow season (average temporal correlation for both years and polarizations). The correspondence may increase by applying a better calibration procedure, for instance taking into account the gain variation throughout the beam. This will be investigated in future work, as for the present experiments, the antenna pattern of the horns has not been measured with sufficient detail to allow for the calculation of an illumination integral as in Strozzi (1996) or Tassoudji et al. (1989). However, since we are mostly interested in temporal variations, this is not a critical limitation for the current study.

At the beginning of our measurements in 2021-2022, a disintegrating ice crust at the bottom of the snowpack may have caused the total tower backscatter to gradually decrease (see also Figure 6). A similar backscatter decrease due to the relaxation of crust features was observed by Lemmetyinen et al. (2016). After the crust disintegration, the backscatter increases both for S1 and the tower measurements, likely due to increasing snow volume scattering until the maximum SD (141 cm) is reached on January 5th. Until the start of the melt season, SD and SWE stayed mostly constant; however, $\sigma_0$ continued to increase due to snow structural changes, especially from February 12th onwards. For example, Figure 6 shows how the scattering in the snowpack increased around February 12th, following a sequence of melt-freeze cycles. In general, for both the tower-based radar and S1, the backscatter (VV and VH) increased by several dB from the start of the snow season until the end of February, which can be attributed to a combination of accumulating snow and changing snow stratigraphy, in part due to mid-winter melt/freeze cycles. For a maximum SD of around 141 cm, the S1 $\gamma_{0,\mathrm{VH}}$ increased by $\sim$3.5 dB throughout the dry snow season, while the tower-based $\sigma_{0,\mathrm{VH}}$ increased by $\sim$7 dB.

During the 2022-2023 season the tower-based $\sigma_0$ fluctuates more strongly over the season than the S1 $\gamma_0$. Figure 9b shows the integrated values at hourly timesteps until February 3rd and daily timesteps afterward, thus the beginning of the season shows the full daily variability rather than the 6 am and 6 pm timeseries in panel a. The cross-pol backscatter slightly increased from the beginning of snow cover until the end of the dry snow season in April. The increase is especially clear from the end of January until peak SD ($\sim$290 cm) at the end of March. The tower-based radar time domain profiles in Figure 8 demonstrate that this increase is likely a result of increasing snow volume scattering. In contrast, the co-pol backscatter fluctuates around a more stable value. The melt season can clearly be separated from the dry snow signal. The tower radar measurements sometimes deviate from the S1 signal. This is especially the case for the tower based HH-polarization that we are here comparing to S1 based VV. Additionally, since tower based radars with a small footprint were used (particularly in the 2022-2023 season), the measurements could be more susceptible to speckle noise, or an incomplete capturing of important processes in comparison with the much larger S1 footprint, which we suggest as a topic for future research.



The results here indicate slightly to moderately increasing values of tower and S1 backscatter throughout the snow accumulation season, at both polarizations during the first year and mostly at cross-pol during the second year. This effect is caused by a combination of a growing snowpack and changes in the snow stratigraphy. Our results are in agreement with the ground based C-band experiments by Kendra and Ulaby (1998), in which an increase in backscatter with SD was observed for an artificial snowpack, especially at cross-pol. In a separate tower-based radar experiment, Strozzi and Mätzler (1998) found that snow scattering was negligible compared to ground reflections. These discrepancies can potentially be explained by different radar systems and setups. In the comparison of the setups used for this current work (Section 3.1), we found that the use of different antennas may have an impact on the results. Furthermore, differences in measurement principles (e.g. integration time, with or without including multiple scattering) and processing choices can also partly explain the variable results in the literature.

## 4    Conclusions

In this paper we presented a unique dataset collected by a tower-based C-band radar system over an alpine snowpack in Idaho. The purpose of these experiments was to improve the conceptual understanding of the interactions between radar waves and the snowpack. These insights can by applied to unravel the potential and uncertainties in S1 SD retrievals. To achieve this, we studied the time-domain backscatter response, which allowed to characterize where the main reflections originated in the profile. The results indicate that at C-band, volumetric scattering happens throughout the dry snow volume. The observed tower-based backscatter increases over time with an increase in SD, but also with the ageing of the snow and melt-freeze induced metamorphosis. The formation of ice features and crusts has a strong effect on the observed backscatter, even when SD and SWE remained constant. This is expected to reduce the performance of S1 SD retrievals in periods and areas of frequent freeze-thaw, when and where such features occur. However, making use of the cross-pol ratio as in Lievens et al. (2022) could partially counter the impact of stratigraphy and microstructural changes, given that the impacts are both manifest in the cross- and co-polarized observations. During the wet snow period, scattering from within the snowpack reduces, the ground surface disappears and the return is dominated by the air-snow interface and the top of the snowpack. Lievens et al. (2022) introduced a method to mask such wet snow conditions. The tower data confirm that under wet snow conditions there is no physical basis for an amplitude based snow depth retrieval. Moreover, it showed that wet snow conditions can indeed be identified by a drop in backscatter relative to the previous dry snow value.

The Lievens et al. (2019) work showed good agreement with SD under various conditions. However, our study was limited to a specific site and to two snow seasons; therefore, repeating the experiment over a longer period and in different snow climates will be helpful to further assess the potential and limitations of using S1 for SD estimation in a wider range of conditions. Furthermore, to properly relate S1 observations to tower-based measurements, the influence of different observing systems and footprint sizes on the collected measurements requires further investigation. A better calibration taking into account the variability in radar gain is recommended. Additionally, a quantitative analysis comparing the contributions of the snow and ground surface and how they evolve throughout the season would yield valuable insights. A side by side comparison with X- and Ku-band measurements would further help to asses the differences and similarities in scattering mechanisms, and



their relative strengths and weaknesses for use in retrieval methods. The advancement of radiative transfer models is another
470 promising pathway to improve the understanding of observed backscatter patterns.

Importantly, this work proves that the snowpack is undoubtedly not transparent to C-band radar waves. Depending on the snow properties, snow volume scattering can even be of similar magnitude as ground-surface scattering. The impact of snow stratigraphy on the backscatter signal may complicate the use of C-band data for satellite-based snow depth retrieval. However, the impact of snow stratigraphy and microstructure changes are anticipated to be potentially even more pronounced for shorter
wavelengths (Kendra and Ulaby, 1998). Over the dry snow period, volume scattering is present at C-band, and satellite-based SD retrievals could perform well.

*Code and data availability.* The tower radar data and example codes for their processing will be made available on zenodo and Github after acceptance of the paper.

*Author contributions.* HPM, IB and HL contributed to the construction and maintenance of the radar system and to the field campaigns. IB
implemented the data processing. GDL contributed to the tower design and to funding acquisition. All authors contributed to the analysis, data interpretation and writing of the manuscript.

*Competing interests.* No competing interests are present.

*Acknowledgements.* This research was supported by the BELSPO project C-SNOW (contract SR/01/375). I. Brangers was funded through the Research Foundation Flanders (FWO). Sentinel-1A/B data are from the ESA and Copernicus Sentinel Satellites project and were down-
485 loaded from the Alaska Satellite Facility and processed using the ESA Sentinel Application Platform (SNAP) version 8. The computer resources and services used for Sentinel-1 data processing were provided by the Flemish Supercomputer Center (VSC), funded by FWO and the Flemish Government, and KU Leuven fund C1 (C14/21/057). We are grateful to Bert Cox and Jona Cappelle (KULeuven) for their technical support with the radars. Multiple people from the Boise State University CryoGARS group have contributed to the collection of field measurements. With special thanks to Megan Mason, Allison Vincent, Thomas Otheim, Thomas Vanderweide, Tate Meehan
and Gabrielle Antonioli. H.P. Marshall and others from the CryoGARS group were jointly supported by the BELSPO project, in addition to NASA Terrestrial Hydrology Program (#80NSSC18K0955) and the U.S. Army Cold Regions Research and Engineering Lab (CRREL, #W913E520C0017).



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
