# Peer review of "Tower-based C-band radar measurements of an alpine snowpack"

_EGUsphere, 2023_

## Referee Comment (RC1)

Snow Water Equivalent (SWE) is a key parameter in hydrological, climatological, and meteorological applications. New efforts for spaceborne radar-based SWE retrieval algorithms like the Lievens et al. (2022) method is promising for alpine snowpack. This paper presents a unique dataset of a tower-based C-band radar system experiment over an alpine snowpack. This dataset is unique and build on the understanding of the Lievens et al. (2019-2022) method. The paper is well presented, and results improved the conceptual understanding of the interactions between the radar waves and the snowpack at C-band.

I have mostly minor comments.

However, I think another analysis could be done to improve the understanding, but I think I will leave it to the authors to decide if they want to add it to the manuscript. This work is enough for publication. Here is what I'm proposing.

Like it is mentioned in the conclusion… Strozzi and Mätzler (1998) found that snow scattering was negligible compared to ground reflections. This brings the question of why the method is working? In a thesis by Jiyue Zhu (2021), they proposed a hypothesis based on radiative transfer modelling for why C-band in cross-pol is sensitive to volume scattering from deep snowpack. In figure 4.11, they show both the volume and background (rough surface scattering) in co and cross-pol as a function of snow depth. At a certain depth, the volume scattering becomes larger than the background scattering for cross-pol where the co-pol always has the background stronger than the volume. Also stated from Zhu (2021) "The volume scattering increases with thickness. Note that, for co-pol, surface scattering is larger than volume scattering. For cross-pol, volume scattering can be larger than surface scattering for large thickness (above 1 meter in the example)." The background contribution in cross-pol is significantly lower than co-pol because the ground does not depolarize the signal. The volume scattering in both co and cross-pol is similar but because the background is smaller, the cross-pol would be more sensitive to volume scattering. Perhaps this dataset would be able to show this.

I suggest, if possible, calculating the backscatter percentage due to snow and ground throughout the season. This would be done with equation 5 and finding an index (i) to separate both intensities (Easier said than done!!). Would be interesting to find the SWE at which the volume becomes stronger than the background. This could strengthen the theoretical hypothesis from Zhu that the Lievens method works.

Minor comments:

Line 64:  I would add that the retrieval is only valid for deep snowpack which differs from previous studies mentioned previously.

Line 179:  typo… should be timestep.

Line 261: The methodology of Lievens et al. (2022) works at lower resolution (100, 500, 1000). Here, the resolution is 5x20. Can you comment if or how this could influence the comparison?

Line 281: I think you can remove "s" in "figures" because it is the same figure. Also change to shows.

Line 350: typo… should be seasons.

Line 426: Can you quantify like you did for 2021-2022 so we can compare?

Line 471-472: This link to my point earlier and it would be great if you can add number to this. How much snow is necessary, so the snow contribution becomes more important than the background.

References:

Jiyue Zhu, Surface and Volume Scattering Model in Microwave Remote Sensing of Snow and Soil Moisture, 2021, Dissertations and Theses (Ph.D. and Master's), DOI https://dx.doi.org/10.7302/3871

---

## Author Response (AR1)

**Below the point by point responses to the reviewers (in blue). These are largely identical to the responses posted on the online discussion platform, with the addition of the relevant changes made to the manuscript and the respective line numbers (referring to the inline version).**

**Reviewer 1**

We would like to thank the reviewer for the constructive feedback to the manuscript.

We very much appreciate the recommendation to link our results with the work by Zhu et al. (2021). This is indeed a very relevant study in the context of this paper, and will be included in the literature section. **Line 57**: "At C-band, DMRT has shown that for deep snowpacks, the magnitude of volume scattering at cross-pol can be larger than the ground surface scattering (Zhu, 2021)."

However, quantifying the relative contributions of snow versus ground may not be feasible given the current setup of the experiments. Even though scattering measurements are collected in the time domain, it is hard to fully separate the ground and snow contributions. The direct volume scatter from the snowpack can be identified, because this portion of the signal is recorded earlier in time than the ground reflections. However, snow will also cause an increase in the multiple scattering, which will be mixed with the ground return signal. Another complicating factor is the wide beamwidth of the antennas. To properly separate the contributions, more research is needed. Therefore, instead of adding this partial incomplete analysis, we would prefer to recommend this as a path for future research. **Line 485-488:** "In addition, calculating the separate soil and snow backscatter would be interesting, yet challenging since the multiple scattering is mixed with the ground signal. Nevertheless, determining the relative snow and soil contributions and how they evolve throughout the season would yield valuable insights and is recommended for future research."

Below the responses to the line-by-line comments.

1. Line 64: I would add that the retrieval is only valid for deep snowpack which differs from previous studies mentioned previously. → This will be added. **Line 67**: "The method works best for deep snowpacks, and was further improved by Lievens et al. (2022)"

2. Line 261: The methodology of Lievens et al. (2022) works at lower resolution (100, 500, 1000). Here, the resolution is 5x20. Can you comment if or how this could influence the comparison? → The resolution used here is 100mx100m (see line 262). There is a trade-off between noise level and the spatial detail. With the Sentinel-1 data we found the noise level of the data to be more stable after averaging to larger pixel sizes. Lievens et al. 2022 also found lower performance at higher resolution, which was mostly attributed to the higher noise levels. We will comment on this in the text. **Line 263**: "This lower resolution was preferred to the original pixel size since it leads to a more stable signal."

3. Line 426: Can you quantify like you did for 2021-2022 so we can compare? → Thank you for this suggestion. We will indeed also add the backscatter changes for the second season. **Line 445-447**: "Specifically, the S1 $\gamma(0,VH)$ increased by ~3 dB, and the tower-based $\sigma(0,VH)$ by ~4 dB during the dry snow season. This excludes the initial spike in tower $\sigma(0,VH)$ at the start of the snow season that is likely caused by soil moisture changes."

4. Line 471-472: This link to my point earlier and it would be great if you can add number to this. How much snow is necessary, so the snow contribution becomes more important than the background. → We appreciate this suggestion, but it is difficult to precisely quantify the magnitudes of the snow and soil scatter contributions, given that multiple scattering cannot be separated from the ground scattering component, and our measurements span a wide range of incidence angles. Furthermore, the snow contribution relative to the background is also dependent on the snow structure. For instance, at the end of the 2021-2022 snow season (Figure 6), the snow contribution is higher than that of the soil; however, in 2022-2023 (Figure 8) the snow contribution, although substantial, does not seem to exceed the strong soil scattering. We will comment on this in the updated text. **Line 485-488** (see earlier point)

**Reviewer 2**

We would like to thank the reviewer for the detailed and constructive feedback to our manuscript.

Below you can find the responses to the individual comments:

1. Even though separating the result and discussion could have been more beneficial for the paper, I think this format is good enough to understand the content fully. More analysis and discussion could have been made on the impact of snow stratigraphy and properties on volume backscattering. The 2022-2023 season is mostly absent from the analysis, even though a whole page of results shows interesting data. → We agree with the reviewer. We initially discussed the 2021-2022 season into more detail, given that more melt-freeze cycles occurred, with a substantial impact on the signal. However, it is true that the 2022-2023 data deserve a more extensive discussion. We will elaborate further on the results of the latter snow season. See additions in manuscript **between lines 341-354**

2. The analysis of the impact of the 2 am recorded time on the season backscatter image could have been more detailed, especially concerning Liquid Water Content. The whole spring diurnal cycle of thawing and freezing is absent in the 2022-2023 season image (figure 8). Did this cause it? → The diurnal cycle of thawing and freezing is indeed not visible in the 22-23 season since (due to technical issues) we only have measurements at 2am for most of the season, instead of hourly as in the 21-22 season. The measurements at 2am only do not facilitate to study diurnal cycles in LWC. Nevertheless we will expand more on the general 22-23 season LWC effects. **Line 406-409**: "Although the data collected at 2 am does not allow for studying diurnal variations in liquid water content, melt and refreeze periods at a longer time scale are visible. The partial refreezing around the 14th of April, for instance, caused a temporary high backscatter right after the onset of melt. For the remainder of the wet snow season, backscatter values are typically low."

3. Also, much effort is put into explaining the differences between the two antenna setups. I would have hoped for more development in the horn antenna results. I did not feel that the authors are concluding if one antenna is better than the other to evaluate which one is best to compare to the Sentinel-1 signal or to evaluate snow parameters, even though the data from the LTSAs antenna are more discussed. → Indeed at the moment we are not able to conclude that one antenna would be better (or more comparable to Sentinel-1) than the other. We think this is still an important outstanding research question. The choice of the antennas does have an impact on the measured signal. For instance, a setup with a more narrow beam (e.g., the horn antennas), sends less energy to near normal incidence angle, reduces the footprint size and might capture less of the volume and multiple scattering. Evaluating this was not the original goal of this work, but rather a question that emerged when trying to understand the differences between the seasons, and will be commented on in the updated manuscript. **Line 369-374**: "Multiple scattering becomes more important as the footprint size approaches the

mean-free-path of scattered photons (Battaglia et al., 2010). In the case of the horn antennas, the narrow beam reduces the footprint size and might capture less of the multiple scattering. The impacts of radar footprint and beamwidth on the received backscatter signal requires more research and will be the subject of future studies. This should include an analysis on which type of antenna would result in a signal that is more comparable to observations from space-born satellites such as S1

4. That being said, this article is well-written. The results come from recent snow seasons, and I understand that this is the first evaluation of this dataset. I am looking forward to seeing a long-term investigation into different frequency FMCW radar results of alpine snowpacks from tower-based instruments. This article will put the ground base for new FMCW radar installation and evaluation for other labs. → We would like to thank the reviewer for this supportive statement. We are also looking forward to seeing more of this type of data and analysis. Note that our experiments were not conducted with an FMCW radar setup, but with an impulse radar system with a wide bandwidth, operating in the time domain. We will more explicitly mention this in the manuscript to avoid confusion. **Line 116**: "Note that we are using an impulse radar system that sends out short pulses, as opposed to a continuous wave used by Frequency Modulated Continuous Wave (FMCW) radars."

Specific comments:

5. Uniformized tower-based and tower based → OK
6. line 45: "Some of the aforementioned tower based papers" I only see Strozzi and Mätler (1998) as a cited and described tower-based paper. The sentence implies multiple citations → More ground/tower based radar experiments are mentioned on Line 38-45. (Lin et al. 2016; Kendra and Ulaby, 1998).

7. line 46 : "side-looking time-domain profiles" I think that side-looking is too general and should be defined clearly. → Side-looking was further clarified. **Line 45-47**: "However, using side-looking time-domain profiles, here at 40° incidence, to help understand scattering processes impacting SAR satellites has not been done previously"

8. line 60: "mentioned tower-based radar studies". Please cite those studies or be more specific. When I read back, I only saw Strozzi and Mätler (1998), cited as a tower-based study. → This also refers to line 41: "Kendra and Ulaby (1998) evaluated model and experimental C- and X-band data of artificial snow and found increasing backscatter with SD at both frequencies". 'Experimental' will be specified more clearly to avoid confusion. **Line 41**: "Kendra and Ulaby (1998) evaluated modelled and experimentally measured, ground-based C- and X-band data..."

9. Figure 5. I am unsure of the necessity of showing the the b) image. I think it lacks context and doesn't bring useful information. → Figure 5 supports the data processing flow, but indeed does not contain essential scientific information. It will be left out in the updated manuscript.

10. line 150 "The first part of the signal..." Can you be more specific? → this refers to the first ~500 samples of the signal. We will add this in the text. **Line 153**: "The first part of the signal, the first ~500 samples, ..."

11. Figures 6 and 8 lack units for the color bar (dB). → will be added

12. Figure 6 and 8. You specified in the text that the permittivity of snow influences the evaluated distance by the radar and is also affected by the ground. If you did not account for the refractive index in the signal evaluation, I would change "Vertical distance (m)" for raw Vertical distance (m) and then define it. → We did take refraction into account (see methods section line 176-182)

13. line 294 Use liquid water content instead of "water content" → OK

14. figure 7 Different markers would help to identify the dates of the profiles → OK

15. figure 7 c) The solid line is not identify → This will be added. Thank you for noting.

16. line 310 (and figure 7) I would have guessed that at least some melt forms would have been present. We clearly see that the snowpack is isothermal and the snow height is dropping. → We agree that it seems as if melt forms could have been misidentified as rounds towards the end of the season. This requires an extra note in the text. **Line 315-316**: "However, the recorded stratigraphy profile at this time does not indicate the grain type as melt forms (Figure 7). The grains were, most likely mistakenly, recorded as rounds"

17. line 327 "The snow-free spring backscatter signal indeed changes with soil moisture fluctuations" Please clearly identify the dates concerned by this phenomenon in figure 6 → The dates will be added for clarity. (The most significant snow-free SM variations occurred on the 5$^{th}$ of June and 12$^{th}$ of June.) **Line 334**: "... changes with soil moisture fluctuations, e.g. on the 5th and 12th of June"

18. Figure 9. It would be easier to understand the figure if the gray shadings were integrated in the legends → This will be added.